Detection of genomic regions associated malformations in newborn piglets: a machine-learning approach

Bakoev Siroj siroj1@yandex.ru 1 2
Traspov Aleksei 1 2
Getmantseva Lyubov ilonaluba@mail.ru 1
Belous Anna 1
Karpushkina Tatiana 1
Kostyunina Olga 1
Usatov Alexander 3
Tatarinova Tatiana V. 4 5 6 7
1 Federal Research Center for Animal Husbandry named after Academy Member LK. Ernst , Dubrovitsy , Russia
2 Centre for Strategic Planning and Management of Biomedical Health Risks , Moscow , Russia
3 South Federal University , Rostov-on-Don , Russia
4 Department of Biology, University of La Verne , La Verne , CA , United States of America
5 Institute for Information Transmission Problems, Russian Academy of Sciences , Moscow , Russia
6 Vavilov Institute for General Genetics , Moscow , Russia
7 School of Fundamental Biology and Biotechnology, Siberian Federal University , Krasnoyarsk , Russia
Piccolo Stephen
Electronic publication date: 2021 Jul 22
Publication date: 2021
Volume: 9
Electronic Location ID: e11580
Received 2021 Jan 10; Accepted 2021 May 19
Copyright: ©2021 Bakoev et al.
Copyright year: 2021
Copyright holder: Bakoev et al.
License: This is an open access article distributed under the terms of the Creative Commons Attribution License, which permits unrestricted use, distribution, reproduction and adaptation in any medium and for any purpose provided that it is properly attributed. For attribution, the original author(s), title, publication source (PeerJ) and either DOI or URL of the article must be cited.
License URL: https://creativecommons.org/licenses/by/4.0/

Keywords: Congenital malformations, Machine learning, GWAS, Agriculture, Pigs

Funding: RSF Project 18-76-00034 Russian Foundation for Basic Research 19-016-00068 A State task of the Ministry of science and higher education 0445- 2021-0008 This research was supported by the RSF Project No. 18-76-00034 (Identification of genome regions related to malformations in newborn piglets); the Russian Foundation for Basic Research 19-016-00068 A (Machine learning-methods for detection of genomic regions and genes affecting piglet splay leg); the State task of the Ministry of science and higher education 0445- 2021-0008 (Genotyping of piglets using GeneSeek Genomic Profiler (GGP) BeadChip for Porcine HD). The funders had no role in study design, data collection and analysis, decision to publish, or preparation of the manuscript.

==============================
Background

A significant proportion of perinatal losses in pigs occurs due to congenital malformations. The purpose of this study is the identification of genomic loci associated with fetal malformations in piglets.

Methods

The malformations were divided into two groups: associated with limb defects (piglet splay leg) and associated with other congenital anomalies found in newborn piglets. 148 Landrace and 170 Large White piglets were selected for the study. A genome-wide association study based on the gradient boosting machine algorithm was performed to identify markers associated with congenital anomalies and piglet splay leg.

Results

Forty-nine SNPs (23 SNPs in Landrace pigs and 26 SNPs in Large White) were associated with congenital anomalies, 22 of which were localized in genes. A total of 156 SNPs (28 SNPs in Landrace; 128 in Large White) were identified for piglet splay leg, of which 79 SNPs were localized in genes. We have demonstrated that the gradient boosting machine algorithm can identify SNPs and their combinations associated with significant selection indicators of studied malformations and productive characteristics.

Data availability

Genotyping and phenotyping data are available at http://www.compubioverne.group/data-and-software/.

Introduction

The mortality of piglets represents a severe loss for the pig farming industry (Pig Progress, 2014; Walters, 2016). The losses predominantly occur during the last days of pregnancy, delivery, and the first three days of life (Woollen, 1993). Many factors contribute to the loss of piglets: the age of sows and their maternal qualities, the birth duration, the litter size, the parity number, farm management, and other factors (Olsson, Botermans & Englund, 2018). Pigs have more frequent congenital malformations than any other domestic animal (Woollen, 1993), and congenital malformations account for a significant proportion of perinatal losses.

The profitable production of pigs requires that a farm consistently produces healthy piglets (Staarvik et al., 2019). However, the piglet mortality rate is positively correlated with the number of piglets born (Olsson, Botermans & Englund, 2018), making it a limiting factor for farm growth. Therefore, identifying and addressing the causes of congenital malformations in pigs is essential for pig farming.

Congenital diseases often pose a diagnostic problem, having unknown etiology, pathology, and prognosis. This is also true beyond animal husbandry: up to 79% of human congenital diseases have unknown etiology (Papatsiros & Others, 2012).

The frequency of occurrence of specific malformations varies from farm to farm. Breeds with small populations have smaller genetic diversity and are prone to more defects. The most common defect, affecting between 2.3% and 5.0% of all newborn piglets, is the piglet splay leg (PSL) (Papatsiros & Others, 2012). PSL is a condition when a newborn piglet cannot hold its hindlimbs together (in severe cases, the forelimbs are also affected) and cannot stand or walk after birth (Partlow et al., 1993) and may, therefore, starve. There are also cases of inguinal, scrotal, and umbilical hernia, cryptorchidism, and atresia. The prevalence of this condition is between 2.3% and 5.0%.

Genetic factors are believed to play an essential role in the development of congenital malformations in piglets. Studies found genetic drivers of PSL (Hao et al., 2017) and cleft palate (which is a lethal anomaly in pigs) describing chromosomal imbalances in affected offspring (Grahofer et al., 2019). All affected offspring were carriers of partial trisomy of chromosome 14, including the FGFR2 gene associated with various dominant hereditary syndromes of craniofacial dysostosis in humans. Hao et al. (2017) presented seven significant SNPs and four genes related to muscle development, glycogen metabolism, and mitochondrial dynamics, identified as potential candidate genes for PSL. However, only limited research aimed at finding SNPs and candidate genes associated with congenital malformations in pigs has been conducted (Bermingham et al., 2014; Ji et al., 2016; Bakoev et al., 2020).

When analyzing extensive genomic data with traditional parametric models, the problem of a small number of observations and a large number of predictive variables (typically, tens or even hundreds of thousands of SNPs are interrogated in thousands or, sometimes, hundreds of samples) is challenging to solve due to the high dimension or highly correlated data structure (Li et al., 2018). Traditional GWAS analysis focuses on a univariate hypothesis and assumes the presence of independent explanatory variables. These methods suffer significantly from a lack of power and accuracy when dealing with the complexity of multiple interactions or correlations between predictors (e.g., SNP-SNP and SNP- covariate interactions) (Lettre, Lange & Hirschhorn, 2007; Zheng et al., 2007; So & Sham, 2011; Adams, Bello & Dumancas, 2015). Machine learning approaches, and Gradient Boosting Machines (GBM), in particular, are suitable for such complex problems (Li et al., 2018). Various machine learning applications were developed to build a nonparametric regression or classification model from data. One strategy is to build a model from theoretical considerations and then define its parameters based on the observed data. The main idea behind the GBM algorithm is to create new baseline learners that are maximally correlated with the negative gradient of the loss function associated with the entire ensemble. There is a wide choice of existing loss functions that can be adjusted depending on the specific task at hand. This flexibility makes GBMs highly customizable for any specific data-driven task. It introduces much freedom in model design, making it a matter of patience since it takes trial and error to select the most appropriate loss function.

We have conducted a pilot GWAS to identify potential genomic regions and genes associated with congenital diseases and anomalies in newborn piglets. We used two common pig breeds for our study: Landrace and Large White. The Landrace pigs originated from the free-breeding, non-pedigreed stock of swine, the regional landrace native to Denmark. The Large White pig was produced in the 18th century by crossing the large indigenous white pig of North England with the smaller, fatter, white Chinese pig. These breeds have different genetics but share the problems with congenital malformations. We use the gradient boosting machine (GBM) approach since it is flexible and relatively easy to implement. Moreover, GBM has demonstrated considerable success in practical applications and various machine learning and data extraction tasks (Bissacco, Yang & Soatto, 2007; Hutchinson, Liu & Dietterich, 2011; Pittman & Brown, 2011; Johnson & Zhang, 2014). We have evaluated the applicability of GSM for searching for genomic areas and genes associated with congenital anomalies and compared its performance to standard approaches.

Materials & Methods

Samples

We have an ongoing collaboration with pig breeding farms in Russia. The tissue samples were collected at the same breeding farm with a closed breeding system by the farm employees over the last decade. Ear clipping and phenotyping were performed following the standard EU monitoring procedures and guidelines (2010/63/EU) at the farm. The piglets that were assessed as healthy were kept alive, and the affected piglets were culled, and the ear clipping was performed post-mortem. Per 2010/63/EU guidelines, ear clipping and pig slaughter do not require a separate IRB approval or anesthesia. We have collected tissue samples, and phenotypic information on all available piglets at the pig farm determined to have birth defects by specially trained farm employees. We have also collected samples from healthy piglets from the same farm to form control groups.

Among the 148 Landrace piglets, 75 were healthy, 27 had piglet splay legs, and 46 had other congenital anomalies. Among the 170 Large White piglets, 108 were healthy, 38 had piglet splay legs, and 32 had other congenital anomalies. All sampled piglets were female.

Genomic DNA was extracted from the tissue samples using a set of DNA-Extran-2 reagents (NPF Syntol LLC, Russia) per the manufacturer’s instructions. The quantity, quality, and integrity of DNA were evaluated using a Qubit 2.0 desktop fluorometer (Invitrogen”/“Life Technologies, Waltham, MA, USA) and a NanoDrop8000 spectrophotometer (Thermo FisherScientific”, Waltham, MA, USA).

The animals were genotyped using the GeneSeek Genomic Profiler (GGP) BeadChip for Porcine HD (Illumina, Neogen), comprising 68,516 SNPs. According to the recommendations, genotype quality control was performed using Plink 1.9 (Marees et al., 2018). After excluding SNPs with a missing sample frequency >2% and a Hardy–Weinberg equilibrium (HWE) p-value <10−7, 45,218 SNPs for the Landrace pigs and 44,324 SNPs for the Large White pigs were retained for further analysis.

Significant SNPs were identified separately for the two breeds and two congenital disabilities.

Data preprocessing and function selection

The malformations detected in newborn piglets were divided into two groups by assessed qualities: leg defects (piglet splay leg, PSL) and other congenital anomalies (CA). The analysis was carried out using a case-control design (Table 1). The dependent variable was binary: presence (“Yes”) or absence (“No”) of the condition, as evaluated by professional pig breeders.

Table 1 Frequencies of congenital disabilities in breeds.

Breed	Yes	No	
Congenital Anomalies	
Landrace	46	56	
Large White	32	106	
Piglet Splay Leg	
Landrace	27	75	
Large White	30	108	
Notes.

“Yes” defect present

“No” defect absent

For the gene-phenotype association analysis, a manifestation of congenital anomalies in piglets and PSL genotypes was used as a predictor in the model. We used the Plink 1.9 0∕1∕2 encoding for genotypes (Purcell et al., 2007). The data were randomly divided into training (80%) and testing (20%) sets. In each iteration of the training, hyperparameters were set up using a search within a 10-fold cross-checking structure on a random 80% of the training set. The most effective hyperparameters were used to train the model and tested on the test set. The gradient boosting machine (GBM) algorithm was used to select the most significant predictors among SNPs.

All scripts were written in R and executed on the H2O platform (Click et al., 2016).

Population structure analysis

To analyze the population structure, we used two related techniques: singular value decomposition (SVD) using the script from (Porto-Neto et al., 2013) and Principal component analysis (PCA), using command princomp in R. Plots of top 10 principal components were done in R. Admixture analysis was performed using the ADMIXTURE algorithm (Alexander, Novembre & Lange, 2009) for varying the number of components between 2 and 9. Cross-validation errors reported by the ADMIXTURE and GPS algorithms (Elhaik et al., 2014) were used to identify the optimal number of components.

GPS algorithm was used for predicting the provenance of all genotyped individuals. If no geographic information is supplied for the reference samples, GPS identifies the closest reference populations to the tested sample. If geographic information is available, GPS also finds a global position where the individuals with the genotype closest to the tested one live. If there is no geographic information, GPS can find the nearest individuals among the reference dataset. GPS is not suitable to analyze recently admixed individuals. GPS works by calculating the Euclidean distance between the sample’s admixture proportions and the reference dataset.

We also used the genomic relationship matrix (GRM), which is calculated by the formula (VanRaden, 2008) G=ZZ′2∑iNpi1−pi, where pi is the alternative allele frequency in the ith locus. Calculation of matrix Z requires several steps. At first, matrix P, containing (2pi − 1) in each cell, is calculated. It has dimension LxN, where L is the number of SNPs, and N is the number of animals. Next, genotype matrix M is calculated, where genotypes are coded as −1/0/1 (−1 is homozygous in alternative allele, 0 is heterozygous, and +1 is homozygous in reference allele). Matrix Z is the difference between P and M, and Z′ isthe transposition of matrix Z.

SVD is a valuable tool for characterizing the genetic structure to detect and extract small signals even if the data is noisy (Berrar, Dubitzky & Granzow, 2007). We performed the SVD and visualized the relationships between populations using R. SNPRelate (Zheng et al., 2012) was used to perform principal component analysis (PCA), Identity-By-State (IBS) analysis, relationship inference, and hierarchical clustering.

Machine learning analysis

GBM can provide an accurate estimate of the response variable by creating new base learners maximally correlated with the negative gradient of the loss function associated with the entire ensemble. Using the classical quadratic error loss function (L2 norm), the training procedure produces a sequence of error-fitting steps. AUC (”Area Under the receiver operating characteristic Curve”) is used to assess classifier performances: better classifiers have larger values of AUC. AUCPR (Precision-Recall) - Logistic Loss –is a logistic loss function that penalizes the classifier’s confidence in an incorrect response; Gini index was used for evaluating the quality of classification, Mean Per-Class Error is the average value for a class error.

We have selected the following parameters of GBM to minimize the validation error: the number of trees (ntree = 1000), learning rate (learn_rate = 10−3), maximum tree depth (max_depth = 10), and number of cross-checks (nfolds = 10).

We stratified the samples by breed and type of defect. All samples were randomly divided into training (80%) and testing (20%) sets. In each iteration of the training, hyperparameters were set up using a search within a 10-fold cross-checking structure on a random 80% of the training set. The most effective hyperparameters were used to train the model and tested on the test set. GBM algorithm was used to select the most “important” predictors among SNPs. GBM approach uses relative importance to rank the SNPs. At each split in each tree, GBM computes the mean squared error (MSE) for regression (the tree split criterion). GBM averages the MSE improvement (decrease) made by each variable across all the trees that the variable is used. The variables with the largest average decrease in MSE are considered the most important. The values of the relative importance of variables vary from 0 to 1 (Minitab, 2010; Natekin & Knoll, 2013). Because GBM is a collection of individual decision trees, the relative importance value is a measure of the critical variable for evaluating the target variable. We considered all SNPs with positive relative importance to be significant.

GBM algorithm can potentially identify too many SNP with positive relative importance when complex traits are analyzed. To reduce the false-positive rate, functional annotation of SNPs is conducted, intersection with other methods is determined, and only the highest scoring SNPs are considered.

SNPs with positive relative importance were used to generate a list of associated SNPs; their positions and corresponding genes were examined with the Ensembl! browser (Sus scrofa 11.1) (Ensembl genome browser 102, 2020). The functional annotation was performed using the Panther database (PANTHER Gene List Analysis, 2020) and the Variant Effect Predictor (VEP) tool (http://uswest.ensembl.org/Sus_scrofa/Tools/VEP/) (McLaren et al., 2016; Khimsuriya et al., 2019). To find the GWAS studies for the human orthologs, we used the EBI GWAS Catalog (https://www.ebi.ac.uk/gwas/home) and a manual search in the literature for data on their associations with any human or animal traits.

Association analysis with rrBLUP and plink

We compared the performance of GBM with rrBLUP (Endelman, 2011) and Plink (Purcell et al., 2007) approaches.

We used the function GWAS from the rrBLUP R package using flags min.maf =0.05 and P3D =FALSE (equivalent to EMMA with REML) and selected SNPs with marker scores −log10p >2, which is equivalent to the condition of p-value <10−2.

Plink was used with the following options: –all-pheno –allow-no-sex –bfile <...>–model –pheno <..>. This command conducts five tests: the tests offered here are (in addition to the basic allelic test): basic allelic test, Cochran-Armitage trend test, Genotypic test, Dominant gene action test, and Recessive gene action test. We chose SNPs with asymptotic p-value <10−2 in at least one of these five tests.

IRB approval

All procedures were performed under the guidelines approved by the LK. Ernst Federal Research Center for Animal Husbandry (Russia) and with the rules for conducting laboratory research (tests) in the implementation of veterinary control (supervision) approved by Council Decision Eurasian Economic Commission No. 80 (November 10, 2017).

Results

We have conducted admixture analysis of both breeds for K = 2...11. Both breeds appear to be genetically diverse. Both Admixture and PCA (Fig. S1) plots showed that LA and LW pigs belong to two different sample groups. However, for all K (Fig. 1), the admixture profiles of Large White pigs are a mosaic of almost all components, while a smaller number of admixture components represents the Landraces. Therefore, the sampled Landrace pigs were genetically more homogeneous than the Large White pigs. To determine the optimal number of components, we have analyzed cross-validation error and accuracy of clustering.

Figure 1 Admixture plots for K = 2 …11 for Landrace (LA) and Large White (LW) pigs.

As a function of the number of admixture components, the cross-validation error drops to a plateau value at K = 4. Next, we used NbClust (Charrad M., Ghazzali N., Boiteau V., Niknafs A., 2014) kmeans procedure in R to partition the samples using 26 different scores to define the optimal number of clusters. The number of clusters for K>3 was between 6 and 8.

GPS (Elhaik et al., 2014) leave-one-out analysis validated the clustering assignment correctness. For K = 2 and 3, nearly 10% of the samples were misclassified (the rest was placed to correct clusters). For the higher values of K, GPS was placing 100% of samples into correct clusters. The fraction of affected animals per cluster ranged from 11% to 42% for splay leg phenotype and 13% to 83% for other congenital anomalies. For example, for K = 11, in cluster LA_2, all six piglets were affected; one had a splay leg and five other congenital anomalies. In cluster LW_3, having 23 members, three piglets were born with splay legs, and no other anomalies were detected.

We have calculated Pearson’s correlation coefficient between piglets’ status and values of admixture components. For example, for K = 18, in both breeds, the presence of congenital anomalies is negatively correlated with the admixture component K8 (LW: r =−0.26, t-test p-value =0.0002). This effect is especially pronounced in Large White piglets: 3% of piglets with K8>0.3 (1 out of 38) have congenital defects, as compared to 31% of piglets with K8<0.3 (29 out of 94). Cluster LW_7, where K8 is the main ancestry, is the healthiest. Congenital anomalies in Large White piglets are positively correlated with the admixture component K5 (r = 0.21, t-test p-value = 0.007). 41% of piglets with K5>0.3 (15 out of 37) have congenital defects, as compared to 16% of piglets with K5<0.3 (15 out of 95). Therefore, it appears that certain pig lines are more susceptible to the development of congenital anomalies.

Fst between Landrace and Large White pigs was estimated by the method of Weir & Cockerham (1984) to be 0.165. Hierarchical clustering (Fig. 2A), PCA analysis (Fig. 2B) showed the separation of Landrace and Large White pigs into two distinct groups. Note that the PCA plot had the arch and horseshoe effects for Large White pigs. These effects are common in ecological and population studies when the sampling units cover a long ecological gradient. Those samples at each end of the gradient have few species in common. These artifacts produce visually beautiful but false patterns that are not present in the original data. Using the SNPRelate package in R (Zheng et al., 2012), we have calculated relatedness (Fig. 2C) and IBS (Fig. 2D). Within- and between-breed relatedness have been inferred by the IBD using the Plink method of moments (Purcell et al., 2007). We have calculated the Cotterman coefficients of relatedness K0 and K1 (Cotterman, 1940). These plots show that Large White samples mainly were from unrelated individuals, while the Landrace samples range from unrelated to first-degree relatives. Clustering (A) and IBS (D) plots also showed population stratification within the Large White pigs: the samples could be separated into two main clusters.

Next, we used the rrBLUP (Endelman, 2011) R package and Plink (Chang et al., 2015) to find significant SNPs associated with the splay leg and congenital malformations phenotypes separately for the Large White and the Landrace breeds. Plink performs five tests: basic Allelic test, Cochran-Armitage trend test, Genotypic test, Dominant gene action test, and Recessive gene action test.

Using the p-value cut-off of 10−5, there were no significant SNPs in Landrace pigs identified by Plink (p-value <10−5 in at least one of the five Plink tests) and also by rrBLUP (Fig. 3). In Large White pigs, one SNP (MARC0062453 a.k.a. rs80912121) was found significant by two Plink tests for piglet splay leg (its rrBLUP p-value = 1.334 ×10−5, hence, it nearly misses the cut-off). It is an intron variant for CRISP3 (cysteine-rich secretory protein 3) on chromosome 7. For other congenital anomalies of Large White pigs, rrBLUP has identified two SNPs, intergenic variant ALGA0086405 and downstream gene variant WU_10.2_13_216912741, the first of each was significant in four out of five tests, according to Plink. SNP ALGA0086405 (rs81454026) is located on Chromosome 15 position 92,733,414. There are three genes near this SNP: GULP, TFPI, and CALCRL. Gene GULP PTB (domain-containing engulfment adaptor 1) is located on Chromosome 15: 92,838,715–93,251,019. Gene TFPI (tissue factor pathway inhibitor) is located on Chromosome 15: 92,345,075–92,409,905. Gene CALCRL (calcitonin receptor-like receptor) is located on Chromosome 15: 92,233,841–92,332,363. These genes may be associated with the disease since, due to artificial selection and small effective population size in pigs, the LD in pigs is much larger than in humans (Badke et al., 2012).

Figure 2 Diversity of samples analysis.

Hierarchical clustering (A), PCA analysis (B) showed the separation of Landrace and Large White pigs into two distinct groups. Relatedness (C) and IBS (D) plots show that Large White samples mainly were from unrelated individuals, while the Landrace samples range from unrelated to first-degree relatives. Clustering (A) and IBS (D) plots also showed population stratification within the Large White pigs: the samples could be separated into two main clusters.

Figure 3 Manhattan plots.

Manhattan plots for splay leg (A, B, E, F) and other congenital malformations (C, D, G,H) phenotypes in Landrace (A, B, C, D) and Large White (E, F, G, H) piglets, rrBLUP and Plink. Since Plink performs five tests for each SNP, the average negative logarithm of the p-value is plotted for Plink.

SNP WU_10.2_13_216912741 (located downstream of a lncRNA located on Chromosome 13: 207,036,621-207,042,808 reverse strand) was not significant according to Plink. Plink has identified four SNPs (ALGA0097461, ASGA0009945, WU_10.2_2_80100147, and WU_10.2_6_139448200), that were not significant according to rrBLUP. ALGA0097461 is in an intron of a lncRNA on Chromosome 18: 23,230,190–23,234,168, forward strand. ASGA0009945 is an intron variant of gene growth arrest-specific 2 gene GAS2 on chromosome 2: 36,713,106–36,840,407 reverse strand. WU_10.2_2_80100147 is an intron variant of TBC1 domain family member 9B, located on Chromosome 2:78548370 (forward strand). WU_10.2_6_139448200 is an intergenic variant on Chromosome 6:151776796 (forward strand).

Table 2 Model performance statistics.

	LogLoss	Mean Per-Class Error	AUC	AUCPR	Gini index	
LW_CA	0.5540447	0.03846154	0.9805258	0.9668851	0.9610516	
LW_PSL	0.5211852	0.0304878	0.985684	0.9446652	0.971368	
LR_CA	0.680473	0.02591362	0.9920266	0.9922641	0.9840532	
LR_PSL	0.5893216	0.02678571	0.9902597	0.9762245	0.9805195	

Table 3 Significant SNPs for Congenital Anomalies in Landrace and Large White.

SNP	rs	Relative (scaled) importance	Chr	Positiont	Region	Genes	
Landrace	
WU_10.2_11_75539421	rs327546322	1	11	68243122	intron variant	UBAC2	
ASGA0090409	rs81308724	0.17969795	9	132259417	intron variant	–	
ALGA0039990	rs80853265	0.162018797	7	27590000	intron variant	KHDRBS2	
H3GA0012640	rs80952853	0.159128694	4	41057067	intron variant	–	
ASGA0001071	rs80929780	0.153784938	1	11602546	intron variant	TFB1M	
ALGA0021723	rs81378521	0.147428179	3	129009157	intergenic variant	–	
CASI0008977	rs81362278	0.134018265	2	107190210	intergenic variant	–	
MARC0043492	rs81236079	0.130584752	10	57103362	intron variant	PARD3	
WU_10.2_12_54317583	rs334985085	0.093871326	12	52096685	intron variant	–	
WU_10.2_5_100991426	rs332283409	0.086313799	5	96174863	intergenic variant	–	
MARC0032403	rs81226924	0.085383243	6	153229902	intron variant	–	
WU_10.2_12_55815693	rs333979944	0.08394744	12	53340648	upstream gene variant	HES7	
WU_10.2_17_2109978	rs333466948	0.08380046	17	2254601	intron variant	SGCZ	
WU_10.2_3_117349436	rs326261063	0.079718525	3	110594886	intergenic variant	–	
WU_10.2_7_7152107	rs319964590	0.079475487	7	6938938	intergenic variant	–	
WU_10.2_2_149667584	rs318724234	0.060478378	2	143464372	intergenic variant	–	
ALGA0092396	rs81464168	0.009155868	16	78922265	intergenic variant	–	
H3GA0019737	rs80783940	0.008957679	7	5925551	intergenic variant	–	
ALGA0011791	rs81360893	0.008939152	2	10024147	upstream gene variant	LRRC10B	
ALGA0061932	rs81431030	0.008708879	11	37414879	intergenic variant	–	
INRA0001740	rs337471656	0.002194657	1	34468175	intergenic variant	–	
ALGA0019218	rs81371006	0.002145075	3	57271701	intron variant	–	
ALGA0044485	rs80793422	1.36E-07	7	108105220	intergenic variant	–	
Large White	
ALGA0017725	rs81374120	0.546335895	3	14378300	intron variant	AUTS2	
ALGA0073944	rs81442837	0.499235802	13	202632332	intergenic variant	–	
WU_10.2_3_15378840	rs341893226	0.180246643	3	15160891	intron variant	GALNT17	
ASGA0072045	rs81459151	0.145878845	16	5329922	intergenic variant	–	
ALGA0113804	rs81342635	0.144408707	15	3184616	intergenic variant	–	
ASGA0013696	rs81375919	0.099250196	3	15479959	intergenic variant	CALN1	
H3GA0007467	rs81363418	0.094527613	2	117920694	intron variant	YTHDC2	
SIRI0001468	rs336235921	0.086802683	14	16125221	intergenic variant	–	
WU_10.2_9_118577560	rs338897394	0.079731895	9	107810783	intergenic variant	LAMB4	
H3GA0032097	rs80839766	0.057702654	11	61255621	intron variant	GPC5	
WU_10.2_4_141926663	rs336392887	0.042778974	4	129408094	intron variant	–	
ALGA0033179	rs81385720	0.03855201	5	80890102	intergenic variant	–	
ALGA0051763	rs81407215	0.037760121	9	18530451	intron variant	DLG2	
H3GA0006930	rs81359994	0.010964118	2	2254601	intron variant	–	
INRA0028052	rs339217230	0.010576312	7	107218015	intergenic variant	–	
ASGA0097745	rs81317775	0.01055165	1	17142254	upstream gene variant	UST	
WU_10.2_16_79725318	rs325342191	0.004863526	16	73720088	intergenic variant	–	
WU_10.2_7_120970587	rs339280644	0.001789988	7	114362106	downstream gene variant	CHGA	
ALGA0092260	rs81463703	0.001774664	16	76426659	intron variant	ADAMTS16	
WU_10.2_9_41095074	rs342351984	0.00168734	9	36909534	intron variant	EXPH5	
ASGA0103580	rs81303379	2.76E-07	9	21603270	intron variant	RAB38	
INRA0054882	rs343281670	6.96E-08	17	58675609	intron variant	APCDD1L	
ALGA0118274	rs81325152	4.80E-08	9	45104488	intron variant	DSCAML1	
M1GA0007538	rs81385962	4.43E-08	5	11566047	downstream gene variant	–	
ALGA0091619	rs81461997	2.37E-08	16	69786647	intergenic variant	–	
DIAS0002910	rs81218446	1.66E-08	6	19956188	synonymous variant	CFAP20	

Next, we used the GBM approach (see Table 2, Table 3, Table S5, and Table S6. A complete list of GBM SNPs is in Table S7). GBM has identified 205 variants combined from separate analyses of two breeds and two conditions. The Ensembl Variant Effect Prediction tool classified the SNPs into the following categories: intron variant: 71%; non coding transcript variant: 9%; intergenic variant: 8%; downstream gene variant: 5%; upstream gene variant: 4%; 3-prime UTR variant: 2%; non coding transcript exon variant: 1%; synonymous variant: 1%. To compare the GBM and Plink performance, we have calculated the confusion matrices (Table S8). GBM was more accurate than Plink in labeling the affected piglets as affected, but this came at the cost of a higher false-positive rate. After training and testing the models, the primary performance statistics were calculated (Table 2).

GBM has found 49 SNPs (23 SNPs in LA pigs and 26 SNPs in LW) linked with congenital anomalies. Twenty-two of these SNPs were located in genes (Table 3). In total, 156 significant SNPs were identified for the splay leg condition. In Landrace pigs, twenty-eight SNPs were identified; fifteen were located in genes (Table S5). In Large White pigs, 128 SNPs were identified (63 of the SNPs were in fifteen genes, Table S6).

The overlap between GBM, Plink (selecting SNPs significant in at least one of the five Plink tests), and rrBLUP identified SNPs is shown in Figs. S2 and S3. Only five SNPs were detected by all three methods: ALGA0075476, MARC0030657, WU_10.2_6_138715609, CASI0008977, and ALGA0021723 (Fig. 3).

ALGA0075476, significant in Landraces for splay leg, is on the 14, position 12790416 in the intron of homeobox-containing gene 1, HMBOX1. According to the EBI GWAS catalog, SNPs in this gene’s human ortholog are associated with obesity-related traits (https://www.ebi.ac.uk/gwas/variants/snp:rs2221894).

MARC0030657 (also a significant marker in Landraces for the splay leg) is on chromosome 1, position 112210717. This SNP is located in the Intron 5–6 of lncRNA ENSSSCG00000048340. In general, lncRNAs serve as molecular signals to regulate transcription.

WU_10.2_6_138715609 (significant in Large White pigs for the splay leg) is an intergenic variant on chromosome 6, position 150780123. Two more intergenic variants (CASI0008977 at chr2:107190210 and ALGA0021723 chr3:129009157) were identified as significant in Landrace pigs for the splay leg.

SNP WU_10.2_18_48204887 was identified as significant by GBM for the splay leg in Large White pigs, for other congenital anomalies in Landrace pigs by Plink and rrBLUP. WU_10.2_18_48204887 (on chromosome 18, position 43798925) is in the intron of gene TRIL (TLA4 interactor with leucine-rich repeats). In humans, SNPs in this region are associated with anomalies in the urogenital system.

The reason for the small number of SNPs in the overlap between the three methods is in the approaches used by the programs: the first two programs use parametric methods (negatively affected by small sample size and a large number of SNPs), while in the GBM algorithm, a nonparametric approach is taken.

In Landraces, one of the identified genes (ACSL3, Acyl-CoA synthetase long-chain family member 3) encoded a metabolite interconversion enzyme from the long-chain fatty acid coenzymes A-ligases. All members of this family convert free long-chain acids into esters but differ in tissue-specific expression. Complex fatty acyl-COA esters are substrates for numerous fatty acid metabolic pathways, including mitochondrial β-oxidation and phospholipids, and triacylglycerol synthesis. Changes in ACSL enzymes can contribute to various physiological processes, including apoptosis, gene transcription, signaling, and vulnerability to oxidative stress (Johnson et al., 2012).

SNP rs344559304 in BTB16 gene (zinc finger and BTB domain-containing 16 as PLZF and ZFP145) received the top score in PSL in Large While pigs. The protein it encodes belongs to the Krüppel-like family of zinc finger proteins. It has been previously shown that ZBTB16 acts as an essential transcriptional repressor or activator (Kolesnichenko & Vogt, 2011) and participates in diverse biological processes (Dick & Doulatov, 2009), spermatogenesis (Buaas et al., 2004; Costoya et al., 2004), the formation of hind limbs (Barna et al., 2000), and the apoptosis of cells (Cheung et al., 2010). Impressive results were obtained in studies of the effect of inactivated ZBTB16 on the formation of the skeletal structures of the leg (Wei et al., 2018): ZFP145 -/- mice showed morphological defects affecting the hindlimb’s skeletal structures, whereas the forelimb’s skeletal defects occurred with lower frequency. Also, ZBTB16 was shown to regulate the expression of the HOX and BMP genes (Wei et al., 2018), influencing leg defect formation.

For PSL in LA pigs, the top SNP was rs81225364, inside an extended non-coding RNA region. LncRNAs participate in many biological processes, such as the regulation of epigenetic modifications (Roberts, Morris & Weinberg, 2014), embryo development (Zhang et al., 2014), and the development of skeletal muscle (Shi et al., 2020).

We have identified SNP rs327546322, located in the UBAC2 gene (ubiquitin-associated domain-containing protein 2), to be of the highest significance for the congenital anomalies in Landrace pigs. Polymorphisms in the UBAC2 gene are associated with a genetic predisposition to Behcet’s disease in humans (Sawalha et al., 2011; Yamazoe et al., 2017), a rare disorder causing blood vessel inflammation throughout the body: in joints, vascular system, lungs, gastrointestinal tract, central nervous system, and epididymis (Kaklamani, Vaiopoulos & Kaklamanis, 1998). The symptoms include recurrent eye inflammation, oral and genital sores, and skin lesions. No information on polymorphism of the UBAC2 gene in pigs is available in the literature; however, based on the association of the gene with Behcet’s disease and its manifestation in humans, the UBAC2 gene is a promising candidate for further investigation of congenital anomalies in pigs.

For congenital anomalies in LW pigs, the top-scoring SNP is rs81374120 in the AUTS2 gene. AUTS2 is one of 10 genes with the highest intron RNA score in the fetal brain (Sultana et al., 2002; Bedogni et al., 2010; Oksenberg & Ahituv, 2013). (Ameur et al., 2011) showed that alternative splicing programs control the expression of intronic RNAs in the fetal brain. In recent years, the role of the AUTS2 gene in developing autism spectrum disorders has been extensively studied in humans. Also, variants of this gene are associated with brain malformations and congenital disabilities (Beunders et al., 2013). Therefore, it can be hypothesized that the identified SNP rs81374120 in the AUTS2 gene in pigs is associated with nervous system development disorders and leads to congenital anomalies in piglets.

Discussion

Currently, assessing genetic load is one of the primary problems in pig farming. Various anomalies affect pig breeding’s economic efficiency, and the search for genetic factors associated with their manifestation is an urgent task for researchers. It is vital to simultaneously reduce the number of carriers of congenital defects in the Sus scrofa population and improve productivity indicators. It is not possible to solve these problems using classical breeding without using modern genomic selection approaches.

The traditional strategy of culling piglets with phenotypic defects is effective in only a small part of hereditary pathologies caused by dominant or sex-linked genes. However, most Sus scrofa genetic diseases have a recessive or polygenic type of inheritance; therefore, these culling methods are insufficient.

Therefore, the task of finding new methods to identify animals that carry hereditary pathologies is so essential. Nowadays, congenital anomalies cause significant economic losses in the pork industry. The development of genomic technology has given new meaning to the search for solutions.

When analyzing large genomic datasets using conventional parametric models, small observations and large numbers of predictive variables present an obstacle. This problem is challenging to the high dimensionality or strongly correlated data structure (Chen & Ishwaran, 2012). GWAS analysis, based on traditional statistical methods, focuses on a one-dimensional hypothesis and assumes independent explanatory variables; these methods suffer significantly from a lack of power and accuracy when dealing with the complexity of multiple interactions or correlations between predictors (e.g., SNP–SNP and SNP–covariate interactions) (Lettre, Lange & Hirschhorn, 2007; Zheng et al., 2007; So & Sham, 2011; Chen & Ishwaran, 2012; Adams, Bello & Dumancas, 2015).

The etiology and pathogenesis of the splay leg are complex and still poorly understood. Histomorphological studies have described PSL as myofibrillar hypoplasia, but this condition has also been detected in clinically normal piglets (Ducatelle et al., 1986). Ultrastructural analysis has shown that piglets with PSL have higher muscle glycogen content than normal piglets (Antalíková, Horák & Matolín, 1996). Maak et al. (2009) compared gene expression in posterior leg muscles (M. adductores, M. gracilis, M. sartorius) from diseased and healthy piglets. As a result, they proposed four genes (SQSTM1, SSRP1, DDIT4, and MAF) related to splay legs in piglets. Hao et al. (2017), in a study on 185 animals from four populations (Yorkshire, Duroc, Landrace, and crossbred Yorkshire and Landrace), identified four genes (HOMER1 and JMY on SSC2, ITGA1 on SSC16, and RAB32 on SSC1) associated with muscle development, metabolism, and mitochondrial dynamics that were proposed as candidate genes for PSL.

Our work has identified 79 genes that could be considered candidate genes for the splay leg. Even though the genes identified by Hao et al. (2017) were not identified in our work (neither by plink or rrBLUP analysis), it may be noted that in LW pigs, seven of the identified genes (EXT2, NDUFA10, MTMR12, HS6ST3, DBT, ACOXl, CHST11) belong to the protein class “metabolite interconversion enzyme” (PC00262). Metabolic changes have been documented in multifactorial diseases in humans (DeBerardinis & Thompson, 2012). Metabolism is carried out through enzymatically catalyzed biochemical reactions and involves the mutual conversion of small molecules (metabolites) that play a crucial role in various cellular functions, from energy production to complex biosynthesis macromolecules (Pey et al., 2013). The EXT2 gene (Exostosin Glycosyltransferase 2) encodes one of the glycosyltransferases involved in the chain extension stage of heparan sulfate biosynthesis. EXT2 gene polymorphism is associated with humans with exostosis (or osteoma, benign growth of new bone on top of the existing bone) (Wuyts et al., 1998).

Conclusions

Our pilot study showed that machine learning approaches could identify genomic loci associated with congenital malformations in piglets. We have identified lists of SNPs and candidate genes associated with congenital anomalies and piglet splay leg for Large White and Landrace pigs. We have also highlighted the need to study farm animals to determine population characteristics and identify genotypes associated with significant selection indicators of malformations and productive qualities.

The small sample size used here is a limitation of our pilot study. We will extend this approach to larger datasets since it is vital for ensuring efficient and ethical handling of farm animals.

Supplemental Information

Supplemental Information 1 Supplementary figures

(1) PCA (2) Overlap between SNP lists

Click here for additional data file.

Supplemental Information 2 Supplementary tables

Click here for additional data file.

Supplemental Information 3 Phenotype data for all samples

Click here for additional data file.

Supplemental Information 4 Genotyping data for of the piglets used in our study (148 Landrace and 170 Large White samples)

FAM, BIM, and BED files.

Click here for additional data file.

Additional Information and Declarations

Competing Interests

Author Contributions

Animal Ethics

Data Availability

Tatiana Tatarinova is an Academic Editor for PeerJ.

Siroj Bakoev conceived and designed the experiments, performed the experiments, analyzed the data, prepared figures and/or tables, and approved the final draft.

Aleksei Traspov, Lyubov Getmantseva, Anna Belous, Tatiana Karpushkina, Olga Kostyunina and Alexander Usatov conceived and designed the experiments, performed the experiments, prepared figures and/or tables, and approved the final draft.

Tatiana V. Tatarinova analyzed the data, prepared figures and/or tables, authored or reviewed drafts of the paper, and approved the final draft.

The following information was supplied relating to ethical approvals (i.e., approving body and any reference numbers):

Ethical approval is not required for genotyping of farm animals in the Russian Federation.

All procedures were performed under the guidelines approved by the LK. Ernst Federal Research Center for Animal Husbandry (Russia) and with the rules for conducting laboratory research (tests) in the implementation of veterinary control (supervision) approved by Council Decision Eurasian Economic Commission No. 80 (November 10, 2017).

The following information was supplied regarding data availability:

Genotyping and phenotyping data are available in the Supplemental Files and at http://www.compubioverne.group/data-and-software/.

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
