# Peer review of "Detection of genomic regions associated malformations in newborn piglets: a machine-learning approach"

_PeerJ, doi:10.7717/peerj.11580_

## Round 0.1 · original submission · Major Revisions

Reviewer 1 has cited some fundamental flaws in the study design and significance criteria. I would like to give you an opportunity to address these comments, as well as those from the other reviewers. However, it is critical that these be addressed thoroughly.

Reviewer 1 ·

Basic reporting

Paper is correctly written overall. Figures 2 and 3 are not clear.

Experimental design

This experiment is far too small for the purpose stated. It uses two separate populations that are apparently merged, inducing a stratification problem. Analyses by cluster adds some confusion here. How was sampling performed? It looks like incidence of malformations is very high.

Data are not enough, chosen nominal p-values are not stringent at all, p=0.01.

Validity of the findings

See above and below

Additional comments

This paper utilizes a gradient boosting machine to identify SNPs associated with malformations in piglets.

Main issues
- Population size is too small, and malformations may occur due to many independent diseases and therefore genes, assuming diseases is partly under genetic control.
- We do not know whether samples were collected in different farms with different incidences
- Disease is rare and so a case / control study is preferred, ie, collecting a large number of cases. This seems to be done but imperfectly as incidence of diseases is very high.
- The group congenital anomalies CA is a pool of different diseases and genetic causes
- Not clear how cross validation was performed, what % of data were left out for prediction? was it done stratified by breed and class?

Minor
- Quantify 'severe loss' , how important is it?
- Spell out and describe briefly GPS

Reviewer 2 ·

Basic reporting

no comment

Experimental design

no comment

Validity of the findings

no comment

Additional comments

The manuscript detected genomic regions associated with congenital anomalie and piglet splay leg in newborn piglets using a machine-learning approach. The GWAS studies of congenital anomalie and piglet splay leg are interesting in pigs. The machine-learning approach was used to conduct GWAS is interesting. However, the manuscript did not consider the effect of sex to congenital anomalie and piglet splay leg in analysis model because it was reported that sex linked with piglet splay leg. In addition, the manuscript use “SNPs with positive relative importance” as significant criteria to generate top-associated SNPs. Is it feasible? If the number of significant SNPs is large, how to judge the most significant SNPs? The detailed comments list below.
Line 34 change “associated with congenital anomalies” to “associated with congenital anomalies and piglet splay leg”
Line 39 the sentence “complex genotypes associated with significant selection indicators of studied malformations and productive characteristics” is hard to understand. What is complex genotype? What is significant selection indicators?
Line 86 “We use the gradient boosting machine (GBM) approach since it is flexible and relatively easy to implement” The reason is not sufficient for using GBM. Why not use deep learning or random forest?
Line 178 and 182 the criteria value “p-value <10-2” is too low to judge the significance of SNP.
Line Table 1: Congenital Anomalies and Piglet Splay Leg were coded yes and no, respectively. They overlaped, to a certain extent. For examples, in landrace, 56 “no” of Congenital Anomalies contain 27 “yes” of Piglet Splay Leg. How to combine the overlapped part between Congenital Anomalies and Piglet Splay Leg?
Line figure 3: There is only the Manhattan plot of RRBLUP, Please show the Manhattan plot of Plink generated. In addition, please show the genes not SNPs on the Manhattan plot.

Reviewer 3 ·

Basic reporting

no comment

Experimental design

The sample size is not two small, and the case-control design should be clarified.

Validity of the findings

All underlying data have been provided.

Additional comments

using machine learning to detect regions associated with defect traits in pigs is a good option for small sample. The paper is good writing, while I have some comments as follows.

1. L74. Hao et al. (Hao et al., 2017). --- Hao et al. (2017)
2, L108 saved – maintained
3. Figure 2 dots and diamonds are not easy to be identified
4. It is better to divide figure 3 into two figures to clearly illustrate the GWAS results.
5. I am confused the sample size in this study, the authors mentioned 102 Landrace and 132 Large White were collected, while it is not consistent with the number in Table 1. And as to the case-control design, has the control no PSL and other congenital anomalies, are they the common control? Please clarify this as the sample size is very small for GWAS.
6. I suggest the abbreviation of Landrace and Large White be used LL and YY like in most literatures.
7. The authors used the high-density SNP chip to implement GWAS, as the linkage disequilibrium will deduce the dependent among SNPs and bring high false positive, the multiple test is usually taken into account, I suggest the authors to use it in this study.
8. Machine learning and rrBLUP, Plink were implemented to identify regions or SNP related to PSL and other congenital anomalies, could the authors show the overlap of results from these three methods. Because the sample size in this study is too small, it is necessary to discuss the superiority of machine learning in such case.

---

## Round 0.2 · Minor Revisions

Please address the two remaining comments from the reviewers. It would be helpful to address these points in the manuscript itself.

Reviewer 1 ·

Basic reporting

Some issues have been clearly addressed, although the main one, that of a small size, cannot be fixed. At least this should be acknowledged in the manuscript and a word of caution should be advised.

Experimental design

Fine with limitations.

Validity of the findings

Too small.

Reviewer 2 ·

Basic reporting

no comment

Experimental design

no comment

Validity of the findings

no comment

Additional comments

The manuscript make some changes according to the reviewer’s comments. The biggest problem of GBM algorithm is that can give a subsets of SNP with positive relative importance for a arbitrary SNP GWAS. Moreover,usually,the number of SNP with positive relative importance is large. Thus, it is necessary to give a critical value to judge the significance of SNP when the positive relative importance of SNP exceed the critical value. However, the manuscript doesn’t figure out the problem.

---

## Round 0.3 · accepted · Accept

Thank you for addressing the reviewers' concerns.